# Puumala Hantavirus Infections Show Extensive Variation in Clinical Outcome

**DOI:** 10.3390/v15030805

**Published:** 2023-03-22

**Authors:** Antti Vaheri, Teemu Smura, Hanna Vauhkonen, Jussi Hepojoki, Tarja Sironen, Tomas Strandin, Johanna Tietäväinen, Tuula Outinen, Satu Mäkelä, Ilkka Pörsti, Jukka Mustonen

**Affiliations:** 1Department of Virology, Medicum, University of Helsinki, 00290 Helsinki, Finland; 2Faculty of Medicine and Health Technology, Tampere University, 33014 Tampere, Finland; 3Department of Internal Medicine, Tampere University Hospital, 33520 Tampere, Finland

**Keywords:** orthohantavirus, hemorrhagic fever with renal syndrome, HLA, nephropathia epidemica, Puumala hantavirus, biomarker

## Abstract

The clinical outcome of Puumala hantavirus (PUUV) infection shows extensive variation, ranging from inapparent subclinical infection (70–80%) to severe hemorrhagic fever with renal syndrome (HFRS), with about 0.1% of cases being fatal. Most hospitalized patients experience acute kidney injury (AKI), histologically known as acute hemorrhagic tubulointerstitial nephritis. Why this variation? There is no evidence that there would be more virulent and less virulent variants infecting humans, although this has not been extensively studied. Individuals with the human leukocyte antigen (HLA) alleles B*08 and DRB1*0301 are likely to have a severe form of the PUUV infection, and those with B*27 are likely to have a benign clinical course. Other genetic factors, related to the tumor necrosis factor (TNF) gene and the C4A component of the complement system, may be involved. Various autoimmune phenomena and Epstein-Barr virus infection are associated with PUUV infection, but hantavirus-neutralizing antibodies are not associated with lower disease severity in PUUV HFRS. Wide individual differences occur in ocular and central nervous system (CNS) manifestations and in the long-term consequences of nephropathia epidemica (NE). Numerous biomarkers have been detected, and some are clinically used to assess and predict the severity of PUUV infection. A new addition is the plasma glucose concentration associated with the severity of both capillary leakage, thrombocytopenia, inflammation, and AKI in PUUV infection. Our question, “Why this variation?” remains largely unanswered.

## 1. Introduction

Puumala orthohantavirus (PUUV) is an enveloped virus with a tri-segmented negative-strand RNA genome. PUUV is one of the many human-pathogenic members of the *Orthohantavirus* genus. However, in the following, we refer to them using the simple designation “hantaviruses”. They are currently classified into two categories: viruses causing hemorrhagic fever with renal syndrome (HFRS) and viruses causing hantavirus cardiopulmonary syndrome (HCPS), although hemorrhages also occur in HCPS and the heart and lungs are also affected in HFRS. PUUV is the only hantaviral human pathogen in Finland, a country with globally the highest incidence of hantavirus infections and a significant impact on public health. Nephropathia epidemica (NE), caused by PUUV, is also common in Northern Sweden and parts of Central and Eastern Europe [1,2,3].

In PUUV-caused NE, the clinical picture is dominated by acute kidney injury (AKI), which is histologically characterized by acute hemorrhagic tubulointerstitial nephritis. Most of the hospitalized NE patients have AKI, and up to 6% of them require transient dialysis treatment [4]. The case fatality rate is very low, ranging from 0.08% to 0.4% [1,2,3].

The reported average incidence of diagnosed PUUV infections in Finland is 31 to 39 cases per 100,000 inhabitants. From the seroprevalence up to 12.5% in the adult population and the incidence of NE in some areas of Finland, it has been estimated that only 20–30% of infected humans seek medical attention, leading to serological confirmation [1,2,3]. This wide range of clinical outcomes and the large proportion of subclinical cases are a unique feature of human-pathogenic hantaviruses, both among those causing HFRS and HCPS. In the following, we will consider what might lie behind this characteristic of PUUV infection. Is it in the virus or in its human host?

## 2. Role of Different Factors in the Variation

### 2.1. Hantavirus Virulence

The clinical course of human hantavirus infections varies greatly according to the different hantaviruses, ranging from no disease to a mild course with a low case-fatality rate (about 0.1% in PUUV infection) to a severe course up to 40%–50% in Sin Nombre (SNV) and Andes (ANDV) virus infections [2,5].

Genetically and antigenically related hantaviruses can show large differences in virulence. Currently, the Dobrava-Belgrade orthohantavirus species is divided into four distinct European hantavirus genotypes: Dobrava (DOBV), Kurkino (KURV), Saaremaa (SAAV), and Sochi (SOCV) viruses [6]. These genotypes correspond to different phylogenetic lineages and display specific host reservoirs, geographical distribution, and pathogenicity for suckling mice and humans [7]. More detailed studies of these closely related hantavirus genotypes, causing either life-threatening infections (DOBV and SOCV), relatively mild infections (KURV), or possibly only subclinical human infections (SAAV), could reveal the genetic determinants of virus-host interaction mechanisms leading to virulence.

PUUV infection in the reservoir host, the bank vole (*Myodes glareolus*), is subclinical and persistent, with prolonged viremia and virus shedding in saliva and feces [1,8]. The human infection, obtained by inhalation of aerosolized rodent excreta, presents as NE, with varying severity. Experiments in a nonhuman primate (macaque) model of inhalational SARS-CoV-2 virus infection [9] suggest that a large infectious dose may lead to a more severe virus infection. In these experiments, however, the symptoms (fever) developed in less than two days, in contrast to intravenous PUUV infection in macaques, where the first symptoms (loss of appetite) appeared only after a week [10]. Natural human PUUV infection through inhalation of aerosolized rodent excreta has an incubation time of 2–4 weeks [1,2,3,4].

PUUV, as a tri-segmented negative-strand RNA virus, shows rapid genetic microevolution both in vivo in the bank vole and in cell culture, e.g., [11,12]. However, there is no evidence that there would be more virulent and less virulent variants infecting humans [1,2,3,4]. Thus, we propose that the reason for the extensive individual differences in the clinical outcome of PUUV infection resides, at least mainly, in the human host.

### 2.2. Human Genetic Factors and Immunity in the Pathogenesis of PUUV Infection

Immunogenetic investigations have mainly focused on the human leukocyte antigen (HLA) system and genes that encode molecules associated with this complex, such as the C4A component of the complement system. Many associations between alleles, or combinations of alleles, and susceptibility to infectious and autoimmune diseases have been described in humans [13].

In Finland, the individuals with HLA alleles HLA-B*08 and DRB1*0301 are likely to have the most severe form of the PUUV infection with lower blood pressures, higher creatinine [14,15], and more virus excretion into the urine and the blood [15]. On the contrary, individuals with HLA-B*27 have a benign clinical course [16]. In Slovenia, the HLA-DRB1*15 haplotype was more frequent in patients with severe PUUV-HFRS progression than in patients with a mild course of the disease [17]. Similarly, distinct HLA haplotypes are associated with severe Hantaan virus (HTNV) HFRS, severe Sin Nombre (SNV) HCPS, severe Andes virus (ANDV) HCPS, and mild ANDV HCPS [7].

The tumor necrosis factor (TNF) cluster belongs to the class III region of the HLA system [18]. The -G308A SNP in the promoter region of this gene correlates with the severe clinical course of PUUV infection in Finnish patients [19] and is strongly expressed in the kidneys of PUUV-infected humans [20]. The TNF gene is partly involved in severe PUUV disease but is a less important risk factor than the HLA-B*08–HLA-DRB1*0301 haplotype [21].

There is indirect evidence [22] that HLA molecules could act as coreceptors for viruses, but curiously, this potentially important finding has not been confirmed or extended to viruses other than human coronavirus OC43 (HCoV-OC43), which causes the common cold in humans. This report [22] was based on the use of monoclonal antibodies to HLA-A, -B, and -C specificities for blocking HCoV-OC43 infectivity in human rhabdomyosarcoma (RD) cells, transfection of HLA-A3.1 to human plasma cells, and immunoprecipitation with polyclonal antiviral antibodies.

Notably, the PUUV infection risk haplotype HLA-B*08–HLA-DRB1*0301 invariably carries a deletion of the C4A gene encoding the C4A component of the complement system [23], which is important in the pathogenesis of PUUV infection. In addition to vascular leakage, excessive complement activation is in fact a major feature of severe and lethal NE [24,25,26].

Abnormal chest radiographic findings are common in acute PUUV infection. We have found that also the presence and severity of abnormal findings associate with HLA-B*08–HLA-DRB1*0301 and TNF2 alleles [27]. Pleural effusion, as a sign of increased capillary permeability, showed the strongest association with these genetic factors.

Polymorphism of certain cytokine genes may contribute to susceptibility to PUUV infection [28]. Polymorphisms of plasminogen activator inhibitor (PAI-1) and platelet glycoprotein (GP) associate with the severity of AKI and thrombocytopenia [29], and the endothelial nitric oxide synthase G894T polymorphism with the severity of AKI and hypotension [30].

The special nature of PUUV infection may also be evident in the fact that, unlike in HCPS caused by SNV [31], hantavirus-neutralizing antibodies do not associate with lower disease severity in PUUV HFRS [32].

PUUV infection does not cause direct cytopathology (cell death) or apoptosis but induces a vigorous immune response, mediated by both B-cells and T-cells. The PUUV infection has been associated with many autoimmune phenomena and the Epstein-Barr virus (EBV) infection. In early studies [33] we found rheumatoid factors, recognized in viral illnesses, circulating immune complexes, and increased levels of IgM in all 18 patients with PUUV-induced disease; the levels of IgG and IgA increased during the clinical disease but remained normal. We also found anti-endothelial cell antibodies in NE and other viral diseases [34]. We detected autoimmune polyendocrinopathy and hypophysitis after PUUV infection [35]. Human CD8 T-cell response in PUUV infection occurs after the acute phase and, curiously, is associated with boosting of EBV-specific CD8 memory T-cells and transient presence of EBV DNA in some patients, indicative of viral reactivation [36]. These findings may be because EBV is a polyclonal B-cell activator and can also explain the following findings in PUUV infection: nonrelated toxoid immunity in PUUV infection (tetanus-specific and pertussis-specific IgG as well as T-cells) [37], the free immunoglobulin light chains in hantavirus infection [38], and the observed [39] increased risk (73%) for lymphoma (resembling mononucleosis-associated Hodgkin lymphoma) after PUUV HFRS.

Ocular and CNS manifestations in PUUV hantavirus infections show wide individual differences in symptoms such as transient myopia, acute glaucoma, uveitis, headache, nausea/vomiting, dizziness, and PUUV-induced encephalitis [40,41,42,43,44,45,46,47]. A genetic susceptibility to PUUV encephalitis has been suggested in a recent study by Partanen et al. [47]. They found a variant in the toll-like receptor 3 (TLR3) gene that led to compromised receptor activity and abnormal interferon activity. This variant is enriched in the Finnish population, and it is possible that PUUV encephalitis is more common in Finland than in other populations [47]. Hormonal defects are common during PUUV infection and are associated with disease severity and biomarkers of altered hemostasis [48,49,50].

The clinical course of PUUV infection seems to be less severe in children than in adults [51]. The same conclusion was made in a systematic literature review that included 53 published studies. Children with PUUV infections rarely, if ever, need any invasive therapy such as acute dialysis or respiratory support [52]. No significant differences in the clinical severity of NE were observed between HLA-B*08 and HLA-DRB1*0301 positives and negatives [53]. An explanation could be the mild course of the disease in children.

In a Chinese study, it was found that although the incidence of HFRS was higher for males, the incidence of the case fatality rate was slightly higher among females of certain age groups [54]. However, a German study showed that there are no sex-related differences in the severity of PUUV infection [55].

ABO and rhesus blood groups have been reported to affect the risk of COVID-19, hepatitis B, norovirus, and possibly HIV infection. According to our study [56], patients with blood group O may be less susceptible to hypotension, but otherwise, blood groups have no major influences on disease susceptibility or severity during acute PUUV infection.

### 2.3. External Factors and Other Viral Genes Affecting PUUV Infection

In our series of 116 patients with an acute PUUV infection, seroconversion occurred against the Ljungan and lymphocytic choriomeningitis viruses (LCMV), and orthopoxviruses in some patients [57]. Cases with LCMV seroconversions were statistically younger, had milder AKI, and had more severe thrombocytopenia than patients without LCMV. However, the low number of seroconversion cases precludes drawing firm conclusions. Due to the low number of cases, further studies are needed to find out if co-infections with other rodent-borne viruses really have an influence on the clinical picture of PUUV infection.

Smoking is a known risk factor for PUUV infection [58,59,60]. In a large series of NE patients, we have shown that smoking is common in hospital-treated patients and that current smokers suffer from more severe AKI and inflammation than ex-smokers or never-smokers [58]. The putative mechanistic details of the enhanced pathogenesis need further study. Recently, we reported that the alcohol consumption of the patients did not seem to affect the clinical course of acute PUUV infection [61].

The possibility that (re)activation of other viral genes (such as those of chronic infections or integrated viral genomes) affects the clinical outcome of PUUV infection deserves to be studied.

### 2.4. Biomarkers Predicting the Outcome of PUUV Infection, cr

We have detected and evaluated numerous biomarkers in PUUV HFRS (NE), as recently reviewed [62]. This has been performed with two aims: (i) to understand better the pathogenesis of NE [4] and (ii) to use the biomarkers to predict the disease severity and long-term consequences of the disease [63,64,65,66]. As summarized in our recent review [67], cardiovascular, nephrological, and endocrinological consequences have been found in some patients, but severe complications are rare.

The list of biomarkers evaluated is long: interleukin-6, C-reactive protein (CRP), pentraxin-3, indoleamine 2-3-dioxygenase, cell-free DNA, soluble urokinase-type plasminogen activator, resistin, YKL-40, galectin-3 binding protein (Gal-3BP; Mac-2 binding protein), GATA-3, neutrophil gelatinase-associated lipocalin (NGAL), and procalcitonin [62].

In clinical practice, however, only thrombocytopenia, low blood pressure, creatinine, and CRP are used to evaluate the severity, but increasing knowledge about the biomarkers has elucidated the pathogenesis of NE. High plasma interleukin-6 levels are associated with clinically severe disease and can be used as a marker of severity [68]. High CRP as such does not indicate a severe form of NE, and, interestingly, high CRP may even be a protective factor for severe AKI in NE. Interestingly, CRP injection has been found protective in experimental immune-complex-mediated nephritis [69]. High levels of Gal-3BP have been detected in the acute stage of NE [70]. Furthermore, the levels correlate with disease severity, i.e., variables reflecting fluid retention and the length of hospital stay. Interestingly, the Gal-3BP level also correlates with increased complement activation [70]. These results suggest that Gal-3BP could possess antiviral action by triggering an innate immune response via activation of the complement system. Additional biomarkers include elevated cerebrospinal fluid neopterin [71] and plasma B-type natriuretic peptide (BNP) [72].

Recent interesting additions to the list of biomarkers are glucosuria [73] and especially the plasma glucose concentration during the first few days after the onset of fever [74], which was associated with the severity of capillary leakage, thrombocytopenia, and inflammation in patients with acute PUUV infection. The effect on AKI was J-shaped. It is possible that a higher plasma glucose concentration is merely a sign of a more severe disease. However, plasma glucose may also influence the pathophysiological process of PUUV infection by damaging the vascular endothelium.

### 2.5. Perspectives

Notably, reinfections do not exist. Thus, hantavirus vaccines are a potential possibility, but no antiviral drugs or vaccines acceptable by “western” standards are currently used.

However, measures to counteract key events in pathobiology offer a promising strategy. Vascular leakage of endothelial cells (increased capillary permeability), not cell lysis or apoptosis, is a key element in the pathobiology of hantavirus disease. We have described the successful use of icatibant in two very severe cases of NE [75,76,77]. Icatibant is a synthetic polypeptide that acts as a selective antagonist for the bradykinin (BK) type 2 receptor, and it reduces increased vascular permeability and inhibits vasodilatation. BK is a potent vasodilator generated locally by endothelial cells from HMW kininogen by the kallikrein-kinin proteolytic system. BK is a nonapeptide (Arg-Pro-Pro-Gly-Phe-Ser-Pro-Phe-Arg) rapidly degraded by angiotensin-converting enzyme and carboxypeptidase and slowly by aminopeptidase. BK receptor 2 is constitutively expressed and participates in BK’s vasodilatory role. Icatibant is a nontoxic drug licensed for the treatment of acute episodes of hereditary angioedema, which can be life-threatening when swelling compromises the airways. Whether icatibant is useful in other hemorrhagic fevers remains to be seen. Interestingly, increased vascular permeability during experimental hantavirus infection can be prevented with drugs that block BK binding [78]. Icantibant, in fact, may be a useful therapy in various infections since adding it to standard care improved both COVID-19 pneumonia and mortality in a very recent proof-of-concept study [79].

## 3. Concluding Remarks

We conclude that the initial question “Why are the outcomes of a PUUV infection so varying?” remains, at least principally, unanswered. Multiple genetic factors appear to play a role. No differences in the virulence of the PUUV strains have been discovered, but they have also not been thoroughly studied thus far.

The pathogenesis of NE PUUV disease is complicated in many respects. Host immune and inflammatory mechanisms are probably important in the pathogenesis leading to the clinical symptoms. The incubation time is unusually long, and when clinical symptoms appear, the virus has already established a general infection in multiple tissues. This complicates the application of antiviral treatments. However, measures to counteract key events in pathobiology offer a promising strategy, as shown by of the use of icatibant to counteract vascular leakage. Similar approaches are most welcome.

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
