# Peer review of "Puumala Hantavirus Infections Show Extensive Variation in Clinical Outcome"

_viruses, 2023, doi:10.3390/v15030805_

Round 1

Reviewer 1 Report

-It would be more of value if the specific role of each biomarker(inflammatory mediator is clearly identified.

-Only smoking was mention as one of the enviromental factors what about the role of global warming in increasing the number of other host such as the reservoir host, the bank vole (Myodes glareolus) mentioned in your paper.

- Is there some sort of a cytokine storm in AKI and other severe clinical diseases related PUUVA infection as seen in COVID-19.

-Is there a protective HLA ??

-The autoimmune polyendocrinopathy and hypophysitis is it part of the multisystem puuva infection or a post infection sequele please elaborate more.

Author Response

  • Role of each...We have in fact identified that IL-6, CRP, Gal-3BP and Glucose are the most interesting ones. For other see Review [67].  We modified the text.
  • Global warming may move Myodes glareolus more to the north but this will not affect the clinical outcome of individual human infections.
  • Cytokine storm has not been studied, except by histology [20].
  • Protective HLA-B*27 was mentioned (line 90)
  • Autoimmune polyendocrinopathy and hypophysitis: Good question: acute or post-infection sequelae?

Reviewer 2 Report

The manuscript provides an interesting overview regarding the variations of clinical outcome in Puumula hantavirus infection. Regarding the pathogenesis of the nephritis, the authors discuss the potential role of humoral immunity , but in this context, it should be mentioned that nephritis can occur without a humoral immune response as demonstrated by a recent report of hantavirus nephritis in a humoral immunodeficient patient with common variable immundeficiency (CVID) (Steininger et al, Microorganisms 2023).  

Author Response

Thank you ! This is an interesting new article but in our opinion not directly relevant for our article.

Reviewer 3 Report

The manuscript entitled "Puumala hantavirus infections show extensive variation in clinical outcome" is an opinion on extensive clinical variation caused by Puumala hantavirus. The manuscript is voluminous and may create monotonous to readers. The authors may add some graphical presentation to make it more attractive to readers. The Authors should mention a specific aim of this opinion and recommendations for future study regarding the issue as well. However, moderate English correction is required.

Line no. 5: Please full form of "HLA" and write the full form of every abbreviation when using for the first time.

Line no. 15-16: Please check the sentence.

Line no. 15-16: Please check the sentence.

Line no. 18-19: Please check the sentence.

Line no. 166: Age, smoking or alcohol drinking are not environmental factors. Please choose a suitable heading.    

Author Response

  • Recommendations. They are mentioned in the Section 2.5 Perspectives
  • Abbreviations now spelled out.
  • Subtitle Environmental changed to External
  • Age is in fact discussed on lines 151-156
  • Lines 15-18 seem to be OK